# Identification of Key Genes Induced by Different Potassium Levels Provides Insight into the Formation of Fruit Quality in Grapes

**DOI:** 10.3390/ijms24021218

**Published:** 2023-01-07

**Authors:** Hong Huang, Xiaoyan Zhao, Qiao Xiao, Wenjie Hu, Pei Wang, Yuanyou Luo, Hui Xia, Lijin Lin, Xiulan Lv, Dong Liang, Jin Wang

**Affiliations:** College of Horticulture, Sichuan Agricultural University, Chengdu 611130, China

**Keywords:** grapes, fruit color, fruit quality, potassium nutrition, RNA-Seq

## Abstract

Inadequate potassium (K) availability is a common abiotic stress that limits the growth and quality of fruit trees. Few studies have investigated the physiological and molecular responses of grapes at different potassium levels. In this study, an integrated approach was developed for grapevines grown at four different potassium fertilization levels [0 (K0-CK), 150 (K150), 300 (K300), and 450 (K450) g/plant] in combination with metabolite measurements and transcript analysis. The results showed that different K levels affected the accumulation of sugars and anthocyanins in the fruit. At 78 days after bloom (DAB), the K150, K300, and K450 treatments increased soluble sugar content by 37.39%, 31.10% and 32.59%, respectively, and anthocyanin content by 49.78%, 24.10%, and 13.06%, respectively, compared to K0. Weighted gene co-expression network analysis (WGCNA) of DEGs identified a network of 11 grapevines involved. During fruit development, potassium application promoted the accumulation of anthocyanins and sugars in fruit by regulating the up-regulation of *GST*, *AT*, *UFGT* and *SPS*, *HT*, *PK* gene expressions. These results suggest that potassium deficiency inhibits anthocyanin and sugar metabolism. In addition, it promotes the up-regulation of *KUP* expression, which is the main cause of K accumulation in fruits. Together, our data revealed the molecular mechanism in response to different K levels during fruit quality formation and provides the scientific foundation for the improvement of fruit quality by adding K fertilizer.

## 1. Introduction

Grape (*Vitis vinifera* L.) is extremely important to the economies of all countries as a fruit, consumed worldwide as fresh fruit, raisins, and wine [1]. It is becoming more and more popular due to its high nutritional value, good taste, versatility, and high profitability [2]. Worldwide consumption of table grapes has been showing steady improvement [3]. However, how to improve quality has always been a hot topic in viticulture.

Potassium, as a necessary nutrient, plays a significant role in the development of plants, such as enzyme activation, photosynthesis, carbohydrate metabolism, and abiotic stress response [4,5]. Therefore, potassium has an important impact on the yield and quality of grapes. Potassium fertilization significantly increased the soluble solids content of the Sharad Seedless grape, while higher concentrations of potassium treatments increased fruit yield, sugar, and anthocyanin content [6,7]. It was found that K had a significant effect on the sugar accumulation and the increase of the soluble solids in the fruit [8]. The application of potassium treatments can also promote the synthesis of anthocyanins in the peel and improve the coloration of the fruit [9].

There are studies that have reported in detail the physiological mechanisms of potassium in various plants and the molecular pathways concerning the transporters and channels that absorb and mobilize potassium from Arabidopsis root to the organ [10]. In Arabidopsis, there are over 30 membrane proteins that have been found to be specialized in K transport, including *Shaker*, *TPK*, *Kir-like*, *HAK/KUP/KT*, and *KEA*, and they have been found to be important in soil K uptake and long-range K transport.

RNA-Seq is an emerging research technique in transcriptomics and is now widely used to analyze expression profiles, analyze splice variants, identify differentially expressed genes, etc. [11]. RNA-Seq is now widely used in plant functional genomics and has become an important tool for plant functional genomics research [12]. There are also many studies on transcriptomics in grapes, and one study using a transcriptomic approach found that transient heat treatment could regulate the expression of malic acid and anthocyanin synthesis-related genes in green and ripe grapes, which facilitated sugar accumulation during the trans-color period [13].Grapevine requires an abundance of potassium nutrients throughout its life cycle [4]. However, grape fertilization usually fails to meet K requirements, leading to improper K supply in many orchards [14]. Therefore, it is very important to investigate the mechanism of molecular response of grape to different K reactions. Therefore, in this study, we analyzed the composition and content of metabolites related to fruit quality in ‘summer black’ grapes, integrated metabolites and transcriptomes using homologue alignment and WGCNA methods for comparative genomic analysis, mined candidate genes related to grape fruit quality, constructed transcriptional regulatory networks, and provided important information for subsequent functional validation of related genes. This will provide important information for the subsequent functional validation of the relevant genes and also lay the foundation for breeding and quality control of grape fruit quality. This study will not only reveal the theoretical mechanism of potassium application to improve fruit quality, but it also has important practical implications for the rational application of potassium fertilizer to fruit trees, and it can provide guidance for efficient and high quality production of grapes. 

### 1.1. Plant Materials and Treatments

The experiment was conducted on 8-year-old ‘summer black’ grape colonized in Sichuan Agricultural University farm (30°55′ N, 103°64′ E); the altitude is about 530 m, and the average annual temperature is 15.9 °C. The soil of the trial garden is sandy loam, the grapes are shaped in ‘Y’ shape, and double film cover is used to avoid rain cultivation. The potassium treatment applied in this test ditch is potassium sulfate, K_2_O ≥ 52%, produced in Nuoshifu (Weifang, Shandong, China) Agrochemical Co., Ltd.

In addition to other normal managements, four potassium fertilization levels [0 (K0-CK), 150 (K150) (low K), 300 (K300) (low K), and 450 (K450) (high K) g per vine]. In the randomized block experiment, every three trees were a treatment, and each treatment was repeated three times; three trees were spaced between treatments as protection plants. Potassium sulfate was applied at 50 DAB. Samples were collected from 50 DAB and every 7 days until maturity, and a total of 5 samples were collected. During the sampling process, 50 berries were randomly removed from different locations in different bunches of several vines in each treatment, replicated three times, and brought back to the laboratory in ice boxes. Each treatment was brought back to the laboratory for the determination of physiological indicators, such as soluble solids content, soluble sugar, and titratable acid content. Samples were frozen in liquid nitrogen and kept in the refrigerator at −80 °C until used for RNA-Seq.

### 1.2. Potassium Content in Grape Fruits and Soil

Concentration of K was determined by Liang B et al. [15]. In brief, approximately 0.2g per each sample lyophilized (HNO_3_, AR, 98%), which were digested and analyzed with ICP-AES ICP6300 (Agilent Inc., Santa Clara, CA, USA).

### 1.3. Berry Skin Color Measurement

The color was assessed by a CM-2600d colorimeter (Konica Minolta Ltd., Tokyo, Japan), using the CIElab color system. Two equidistant color measurements were performed near the equator of each fruit using *L*, a*, b**, and color space units. *L** is the luminance (0, black; 100, white), *a** is the red/green value (*a** > 0, red; *a** < 0, green), and *b** is the yellow/blue value (*b** > 0, yellow; *b** < 0 blue).

### 1.4. Fruit Quality Parameters

A digital refractometer (Hybrid PAL-BX/ACID 1; ATAGO, Tokyo, Japan) was used to measure the soluble solids (SSC), and titratable acid (TA) was measured by using a previously reported acid-base titration [16]. Determination of soluble sugar performed with the anthrone colorimetric method [17]. Determination of total anthocyanin content was performed with pH difference spectrophotometry [16].

### 1.5. Glucose and Fructose Content

The concentration of glucose and fructose was measured in accordance with the method of Liu et al. [18]. In short, 1 g of flesh was homogenized with 5.0 ml of cold ethanol (80%) and incubated at 35 °C for 10 minutes. The supernatant after centrifugation and filtration was analyzed using an Agilent 1260 HPLC system (Agilent Technologies). An Innoval NH2 column (4.6 mm × 250 mm, 5 μm, Agela Technologies, Shanghai, China) was installed in the system. The mobile phase was made up of a mixture of acetonitrile: water (80:20, *v*/*v*) at 1 ml·min^−1^.

### 1.6. Tartaric Acid and Malic Acid Content

The extraction and determination of organic acids were carried out in accordance with Zhang et al. [19]. An amount of 1 ml of supernatant was filtered with a 0.22 mm organic filter head and it was determined using the Agilent 1100 Series HPLC system. The column was ZORBAX Eclipse XDBC18 (416 mm × 250 mm, 5 μm) with a mobile phase of 2 mg·ml^−1^ metaphosphate solution at column temperature 25 °C, flow rate 0.5 ml·min^−1^; the injection volume is 5μl, and the detection wavelength is 214 nm.

### 1.7. RNA-Seq Data Analysis

The pulp and peel of each treated grape were mixed and immediately placed in liquid nitrogen; each sample was 3 biological replicates, and the sample RNA extraction and transcriptome sequencing were entrusted to ‘Beijing Nuohe Zhiyuan Technology Co., Ltd.’ in Beijing, China. Total RNA was extracted using the RNA prep Pure polysaccharide polyphenol plant total RNA (TIANGEN biotech, Beijing, China), and the cDNA was reverse-transcribed with the M5 Super plus RT-qPCR kit with the gDNA remover kit (Beijing Polymeric Beauty, Beijing, China). All steps were performed according to the reagent instructions. The cDNA library was obtained by reverse transcription and PCR amplification after RNA detection, and after the library inspection was qualified, it was sequenced on the Illumina platform. After obtaining raw data (raw reads) by sequencing, the original data of the sample was processed to obtain high quality data on quality (clean data), and the sequence comparison with the reference genome was used to obtain data for subsequent analysis.

### 1.8. Functional Classification Enrichment Analyses

Differential expression analysis of samples was performed using the expression difference analysis software DESeq2 by FPKM (Fragments Per Kilobases per Million reads) and |log2fold changes|≥1, and a corrected *p*-value < 0.05 as the screening criteria, screening for differentially expressed genes between samples. DEGs were combined with GO (Gene Ontology), and the KEGG (Kyoto The Encyclopedia of Genes and Genomes database was compared to obtain functional annotation information corresponding to genes.

### 1.9. Construction of Gene Co-Expression Network Module

In order to simplify the genes into co-expression modules, a weighted gene co-expression network analysis (WGCNA) was performed using default settings in R [20]. The FPKM values were normalized to create an adjacency matrix, and then the phenotype data were loaded into a WGCNA package to compute the correlation-based association between gene modules and phenotype. In WGCNA, the adjacency matrix was transformed into a topology overlap matrix. Based on the construction of the network, transcripts with similar expression patterns were grouped into a module, and unique genes from these modules were likewise measured. Genes from all modules were exported into Cytoscape with default settings as described by M.J. Umer et al. [21].

### 1.10. Gene Expression Analysis by Quantitative RT-PCR (RT-qPCR)

Based on the enrichment results and WGCNA analysis, the different expression of genes involved in the metabolic pathways were screened, for example, ion transport and anthocyanin biosynthesis, and 11 key genes were selected for RT-qPCR verification and analysis; all 11 candidate genes were upregulated relative to the K0 treatment group. Primers were designed using Primer BLAST (https://www.ncbi.nlm.nih.gov/tools/primer-blast (accessed on 14 December 2022)) using *VvGAPDH* as the reference gene (Table 1). The cDNA was reverse-transcribed with the M5 Super plus RT-qPCR kit with the gDNA remover kit (Beijing Polymeric Beauty, Beijing, China), and RT-qPCR was reversed using the 2xM5 HiPer SYBR Premix EsTaq (with Tli RNaseH) Kit (Beijing Polymeric Beauty, Beijing, China). The thermal cycle program for qPCR is 95 °C 30 s, followed by 40 cycles, 95 °C 5 s, 60 °C 30 s. *VvGAPDH* was used as the reference gene. Analysis of relative gene expression levels used a comparative threshold period (CT) method (2^−△△CT^) [22].

### 1.11. Data Statistics and Analysis

Duncan’s method was used to test the significance using SPSS 27.0. Figures were elaborated using SigmaPlot 12.5 and OriginPro 2023 (64-bit learning version).

## 2. Results

### 2.1. Grape Fruit Potassium Content

The potassium content of fruit was on the rise during the whole development period, and the effect of potassium treatments at different concentrations on potassium accumulation in fruit was different. Compared with the K0, the concentrations of K150, K300, and K450 increased the potassium content by 11.38%, 15.40%, and 8.37% at 78 DAB, respectively (Table 2).

### 2.2. Soil Potassium Content

The potassium content of the soil under different treatments was different at 78 DAB. The lower potassium content was, the more potassium was transferred to the aboveground part; the higher potassium content was, the less potassium was transferred to the aboveground part. Therefore, it can be explained that the potassium transferred to the aboveground section under the three treatments (K150, K300, and K450) was higher than K0. In addition, the K150 treatment effect was the best (Table 3).

### 2.3. Anthocyanin and Soluble Solids Content

The accumulation of anthocyanins is not only related to the color of the fruit skin, but is also an important indicator of fruit ripeness. The anthocyanin content and SSC under the four treatments showed an upward trend in general. Compared with the K0, the concentrations of K150, K300, and K450 increased the anthocyanin content by 49.78%, 24.10%, and 13.06%, respectively, (Figure 1A) and the soluble solids content by 11.80%, 5.60%, and 4.10% at 78 DAB, respectively (Figure 1B).

### 2.4. Soluble Sugars and TA Content

The content of soluble sugars in the fruit was closely related to the quality and ripeness of the fruit. At 57 DAB, the soluble sugar content of fruit was increased, and it increased rapidly at 71–78 DAB. At 78 DAB, the soluble sugar content of K150 treatment was the highest. Compared with the K0, the concentrations of K150, K300, and K450 increased the soluble sugar content by 37.39%, 31,10%, and 32.59%, respectively (Figure 2A).

At 78 DAB, the fruit under K150 and K300 treatments had the lowest TA content. At 50 DAB, the TA content between each treatment was not significant. At 78 DAB, compared with the K0, the concentrations of K150, K300, and K450 decreased the TA content by 22.06%, 21.84%, and 15.98%, respectively (Figure 2B).

### 2.5. Glucose, Fructose, Tartaric Acid, and Malic Acid Content

Different concentrations of K treatments have a significant impact on the content of glucose, fructose, tartaric acid, and malic acid in the fruits, which can play a role in increasing sugar and reducing acid. The content of glucose and fructose was increased by potassium fertilizer treatments at different concentrations. As the fruit developed, the concentration of sugar components in the fruit gradually increased. Compared with K0, the concentrations of K150, K300, and K450 increased the glucose content by 25.51%, 34.50%, and 19.12%, respectively, (Figure 3A) and the fructose content by 20.76%, 26.27%, and 16.03% at 78 DAB, respectively (Figure 3B).

The content of malic acid and tartaric acid in different K treatments was lower than K0. Compared with K0, the concentrations of K150, K300, and K450 decreased the tartaric acid content by 11.15%, 24.84%, and 29.53%, respectively, (Figure 4A) and the malic acid content by 27.44%, 11.01%, and 16.28% at 78 DAB, respectively (Figure 4B).

### 2.6. Berry Skin Color

The CIElab system is the most widely used and recommended system for color evaluation [23]. *L**, *a**, *b** and CIRG color indices are highly correlated with the color of grape berry, and they have been successfully applied to indicate the color of berry. In general, different concentrations of potassium treatments have an impact on the color of grape fruit: L* value represents brightness—the larger the *L** value, the brighter the color, the smaller the L* value, the darker the color, the closer to purple-black; the higher the *a** value, the higher the red component of the fruit, that is, the more deviation from the purple-black; the *b** value represents the yellow-blue difference index—the larger the positive value, the darker the yellow, the smaller the negative value, the deeper the blue. The test shows that the K150 and K300 treatment color is closer to purple-black (Table 4).

### 2.7. Principal Component Analysis (PCA) and Correlation Analysis

We performed principal component analysis (Figure 5A) as well as correlation analysis for the important qualities of 78DAB fruit (Figure 5B). The eigenvalues and contribution rates of the principal components (PC) are the basis of selecting the PCs; the larger the eigenvalues, the more information the PC contains. In this experiment, PCA was performed with eight measured variables (SSC, TA, anthocyanin, soluble sugar, glucose, fructose, tartaric acid, and malic acid). The contribution of the first PC accounted for 78.90% of the total variation. Soluble solids (SSC), anthocyanins, glucose, fructose, and soluble sugar mainly reflected PC1. These variables are important indicators to evaluate the fruit quality of ‘summer black’. The second PC explained 14.10% of the total variation, and titratable acid (TA), tartaric acid, and malic acid mainly reflected PC2. The results showed that the reproducibility within each sample group was good, and the variability between groups was significant. The sugar as well as anthocyanin content of ‘summer black’ grape fruit were the main source of information for PC2, and exogenous potassium applications associated with fruit sugar as well as anthocyanin concentration were positively correlated.

Correlation analysis showed that exogenous potassium application was significantly and positively correlated with fruit sugars and negatively correlated with fruit acids. Although the direct positive correlation with anthocyanin content was not significant, anthocyanin content was significantly and positively correlated with fruit sugar, so exogenous potassium application could affect the fruit sugar concentration and thus anthocyanin content, enhancing fruit coloration. Thus, it can be concluded that exogenous potassium application had a pond-enhancing and acid-reducing effect on the grapes.

### 2.8. DEG Identification after K Treatment

Transcriptome sequencing of a total of 27 samples (T represents 50DAB fruit, A, B, C, D and E, F, G, H represent K0, K150, K300, and K450 of 64 DAB and 78 DAB, respectively) from four treatments was carried out on the Illumina platform, and the Clean Data of each sample reached more than 6.30 Gb. The Q20 and Q30 sequences accounted for more than 95% (the sequencing error rate <1%), and more than 90%, respectively, and the GC content was greater than 45% (Table 5). Pairwise comparisons of samples processed at the same sampling time identified 5840 and 6497 non-redundant DEGs (Figure 6 and Figure 7).

### 2.9. Functional Classification of DEGs

To analyze the function of genes’ different expressions in response to K at 78DAB, the DEGs were enriched with Gene Ontology (GO). The top 20 GO enrichment entries for abundance are shown (Figure 8). In the classification of biological processes, DEGs are mainly enriched into items such as metabolic processes and cellular processes. In the cell composition, it is mainly enriched into membranes, cells, etc. In molecular functions, it mainly includes catalytic activity, binding, and ion transport activity. The KEGG analysis showed that there was significant enrichment of DEGs in several important biological processes, including galactose metabolism, fructose and mannose metabolism, and flavonoid biosynthesis (Figure 9 and Table 6).

### 2.10. Constructing the Co-Expression Network and Identification of Key Candidates

Co-expression networks were constructed using 31,740 DEGs in WGCNA (Figure 10A). FPKM values for 21 different gene modules were determined by entering the RNA-seq data into the WGCNA module. All of the identified modules were highlighted in various colors. Correlation analysis was carried out to study the relationship between module characteristic genes and phenotypes (anthocyanin, sugar, potassium) (Figure 10B). Among 21 co-expressed gene networks, three had significant correlation with anthocyanin, sugar, and potassium content. The turquoise module contained 8754 genes, the blue module contained 7878 genes, and the yellow module contained 1480 genes. The gene network was visualized using the WGCNA R package, and nodes and edges were calculated for the three modules.

To further explore candidate genes that contribute significantly to the gene network, annotation information for all selected genes was obtained from the Grapevine Reference Genome Annotation Database (Figure 10C–E). This comprehensive technique was able to precisely identify important candidate genes by comparing annotation information of pivotal genes in DEGs and validating the selected 11 genes by RT-qPCR. Among them, there are five genes associated with anthocyanin biosynthesis, three of which are associated with sugar metabolism and three of which are responsible for the accumulation of potassium. The results from RT-qPCR were consistent with RNA-Seq. Finally, seven genes (Turquoise.0870, Turquoise.2680, Turquoise.0050, Turquoise.6060, Turquoise.4750, Turquoise.1730, and Turquoise.0850) from the turquoise module, correlated with anthocyanin and potassium accumulation, were identified as anthocyanin acyltransferase (*AT*), glutathione-S-transferase (*GST*), UDP-glucuronosyltransferase (*UFGT*), and potassium transporter (*KUP*). Three genes from the blue module and one gene from the yellow module (Blue.0200, Blue.1060, Blue.1580 and Yellow.3140) correlated with sugar and potassium transporters were identified as Sucrose-phosphate synthase (*SPS*), Pyruvate kinase (*PK*), and potassium transporter (*KUP*). These alleged main candidates are highlighted in red in the gene pool and selected based on their importance in the module (relation to anthocyanin, sugar, and potassium) and their expression profiles. The 11 major candidate genes predicted from the turquoise, blue, and yellow modules and the annotations are listed in Table 7.

### 2.11. RT-qPCR Analysis of Differentially Expressed Genes

RT-qPCR analysis of 11 differential genes was performed (Figure 11). During the whole fruit development period of grape, compared with 50 DAB, the genes related to potassium transporter, glucose metabolism, and anthocyanin synthesis showed an overall upward trend, indicating that the accumulation of potassium, sugar, and anthocyanin in grape fruits mainly concentrated in the middle and late stage. Up-regulation of *UFGT*, *GST*, *AT*, *SPS*, *PK*, *HT*, and *KUP* genes promoted the accumulation of anthocyanins, sugars, and potassium in fruits at 64 and 78 DAB. K150 and K300 had significant effects on genes related to potassium transporter, sugar accumulation, and anthocyanin synthesis at 78 DAB.

## 3. Discussion

### 3.1. Anthocyanin Synthesis-Related Genes Responses to K Treatment

Flavonoids, especially anthocyanins, are the main pigments that determine the color of the grape fruit. As previously mentioned, anthocyanin accumulation was rapidly induced by K application [8]. Transcriptome analysis showed that the expression of five genes involved in anthocyanin synthesis, *GST*, *AT*, and three *UFGT* genes, were upregulated upon K treatment. Among them, the expression of *UFGT* is essential for anthocyanin biosynthesis [24]. In the present study, the expression levels of five anthocyanin synthesis genes were determined by RT-qPCR (Figure 11A). Interestingly, the transcript levels of all five anthocyanin synthesis genes were significantly increased after K supplementation compared to K0. The increase in anthocyanin biosynthesis gene expression coincided with the increase in anthocyanin content in potassium-treated fruits (Figure 1A), indicating that external K application affected anthocyanin production, both in terms of gene expression and metabolite levels [25]. All five genes identified in this study are related to anthocyanin synthesis, and *GST* is involved in anthocyanin transport. *UFGT* is involved in glycosylation transfer, which transfers *UDP-glucose* to the C3 hydroxyl group of anthocyanins, producing anthocyanin 3-glucoside, which improves the color of flowers or fruits. Anthocyanin acyltransferases modify the unstable anthocyanins [26]. These results suggest that several genes associated with anthocyanin synthesis respond to exogenous potassium application during berry ripening. This study also found that K150 and K300 treatments showed higher expression of anthocyanin synthesis-related genes at 78 DAB, which also indicated that low K induced better expression of anthocyanin synthesis-related genes than high K treatments. However, the regulation of the K-anthocyanin regulatory complex needs further study.

### 3.2. Sugar Metabolism-Related Genes Responses to K Treatment

It is well known that K is essential for the activity of many enzymes in plants [24]. Transcriptional analysis studies of genes in Arabidopsis, rice, and sugarcane during low K stress revealed a higher proportion of genes involved in metabolic processes than the sum of stress-responsive genes, such as the expression of many key enzymes essential for metabolism, those involved in pyruvate, sugar metabolism, and flavonoid biosynthesis [25,26]. Some genes of metabolic processes were identified by our transcriptome data, including the metabolism of starch and sucrose, glucose and mannose metabolism, which were expressed in grape fruits (Table 7). Sugar metabolism in the fruit is especially vigorous during ripening, and the majority of sugar comes from the photosynthesis of the leaves. The process of sugar metabolism has been more thoroughly investigated in previous studies [27]. In this paper, we studied different expression levels of the key genes associated with sugar metabolism of fruit maturation at low and high K state (Figure 11B).

In this study, the expression of some genes related to sugar metabolism were influenced by K treatment. Three sugar-related genes (*SPS*, *PK*, *HT*) were detected in this study. *SPS*, Sucrose phosphate synthase, is the rate-limiting enzyme in sucrose synthesis, and *PK*, Pyruvate kinase, is the rate-limiting enzyme in glycolysis and regulates C/N metabolism in metabolic enzymes [28]. Following chronic K deficiency, pyruvate kinase activity may be inhibited, resulting in a marked decrease in the amount of pyruvate in the cytoplasm, which may inhibit glycolysis and many downstream metabolic processes (TCA cycle) [29]. For grape berry, the accumulation of hexose is mainly from sucrose, which is exported from the source (leaves) and introduced into the fruit tissue cells [30]. During fruit ripening, sugar transporter (*HT*) genes also respond to K treatment. In general, compared with the K0 at 78 DAB, the sugar-related genes at different K levels were relatively up-regulated, increasing the concentration of glucose and fructose in grape fruits. This is in line with previous research findings [8].

### 3.3. Potassium Transporter-Related Genes Responses to K Treatment

It is well known that K deficiency can affect the uptake and accumulation of nutrients and their associated transporters [25], resulting in plant growth retardation [31]. We found that the expression of a number of K transporters were also influenced at both low and high K concentrations compared with K0 at 64 and 78 DAB (Figure 11C). This suggests that the growth and reproduction of grapes were poor in the period of K deficiency and that K transport from grape roots to shoots might have been inhibited. Ma et al. [25] showed that K content in rice roots and shoots was lower than normal K under low K stress, and the expression of *OsHAK* (1, 7, 11) was significantly increased. In contrast to that study, our study focused mainly on grape fruit. This study found that the levels of gene expression and channels of K transporters significantly up-regulated the K treatment compared with K0 (Figure 11C). Interestingly, the gene expression associated with potassium transporters in the low K was higher than high K at 78 DAB. These data are generally consistent with our physiological data (Table 2). The three potassium transporter proteins identified in this study belong to the *HAK/KUP/KT* transporter family; it has been shown that the Shaker channel family and the *HAK/KUP/KT* transporter family, with nine and 13 members in A. thaliana and nine and 27 members in rice, respectively, play a major role in plant K^+^ uptake from the soil and long-distance transport [31,32]. It has been reported that specific K channels were linked to sugar loading and unloading in Vicia faba and maize [33]. These results indicate the importance of potassium transporter proteins for fruit development, and of course, there are many more than these three potassium transporter proteins at play for fruit development; this should be further studied in future experiments.

## 4. Conclusions

The results showed that K content, sugar concentration, and anthocyanin content in grapes were significantly affected by K levels, especially when treated with K150 and K300. We performed RNA-Seq on fruit samples collected from 50, 64, and 78 DAB and analyzed gene expression related to sugar, anthocyanin, and potassium concentrations to determine the molecular mechanisms explaining the different responses of fruit to low and high potassium. Based on the analysis of biological functions, the majority of differentially expressed genes were found in the ‘metabolic pathway’, ‘hormonal signal transmission’, ‘ion transmembrane transport’, and ‘ROS reaction’. Different levels of potassium treatment can promote the accumulation of potassium, sugar, and anthocyanin by inducing the up-regulation of potassium transporters genes (3 *HAK/KUP/KT* genes), sugar metabolism genes (*HT*, *SPS* and *PK* genes), and anthocyanin genes (*GST*, *AT*, and 3 *UFGT* genes), thus improving fruit quality. However, low potassium is better than high potassium. This is the first comprehensive study using transcriptomics to study the response of grape berries to low and high potassium environments. The results of this study will provide a theoretical basis for further research on the regulatory mechanism of some candidate genes, reveal the molecular mechanism of improving fruit quality through potassium addition, and also provide a scientific basis for the rational application of potassium fertilizer by fruit farmers in production; of course, some functional validation is necessary in subsequent studies, which are essential to further elucidate the mechanism of fruit growth in grapes due to K/deficiency conditions of application.

## Figures and Tables

**Figure 1 ijms-24-01218-f001:**
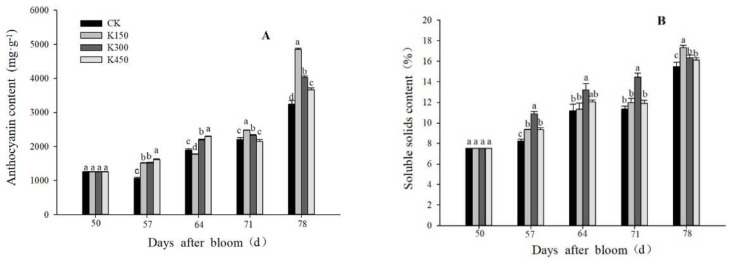
(**A**): Anthocyanin content in skin. (**B**): Soluble solids content in flesh. Different lowercase letters indicate significant differences among the treatments (Duncan’s Multiple Range Test, *p* < 0.05).

**Figure 2 ijms-24-01218-f002:**
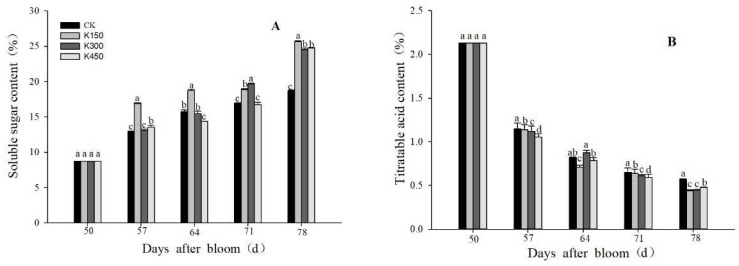
(**A**): Soluble sugar content in flesh. (**B**): Titratable acid content in flesh. Different lowercase letters indicate significant differences among the treatments (Duncan’s Multiple Range Test, *p* < 0.05).

**Figure 3 ijms-24-01218-f003:**
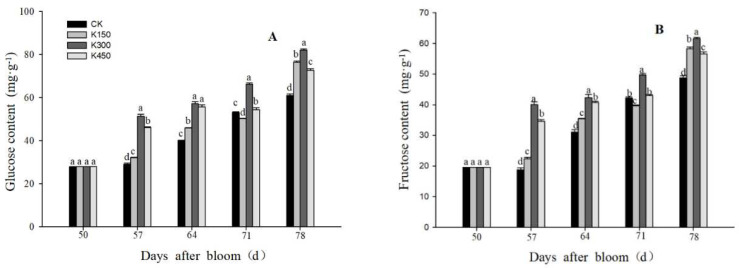
(**A**): Glucose content in flesh. (**B**): Fructose content in flesh. Different lowercase letters indicate significant differences among the treatments (Duncan’s Multiple Range Test, *p* < 0.05).

**Figure 4 ijms-24-01218-f004:**
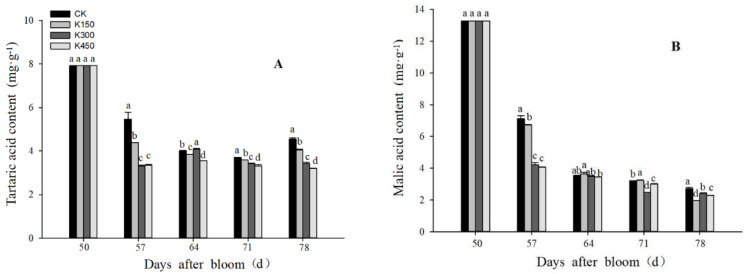
(**A**): Tartaric acid content; (**B**): malic acid content. Different lowercase letters indicate significant differences among the treatments (Duncan’s Multiple Range Test, *p* < 0.05).

**Figure 5 ijms-24-01218-f005:**
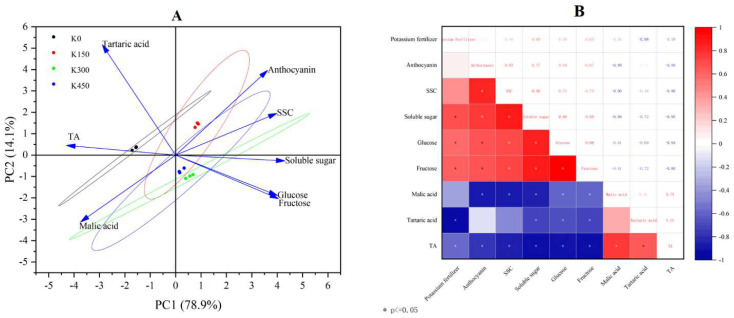
(**A**): Principal component analysis; (**B**): correlation analysis (red represents positive correlation, blue represents negative correlation, the shade of color represents the degree of correlation, and * represents *p* ≤ 0.05).

**Figure 6 ijms-24-01218-f006:**
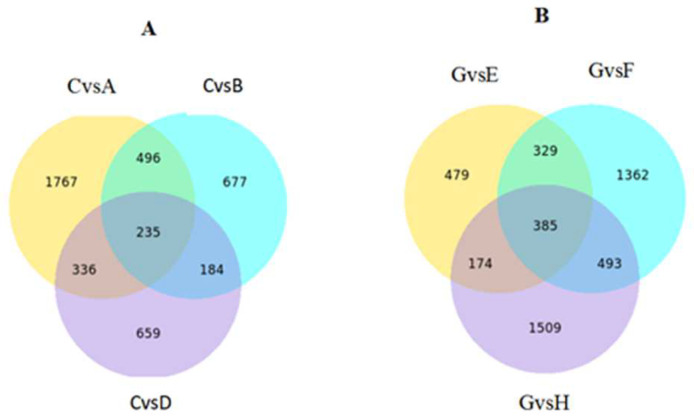
Venn diagram of differentially expressed genes. (**A**): Venn diagram at 64 DAB. (**B**): Venn diagram at 78 DAB. (*p* < 0.05 and fold change equal or greater than 2).

**Figure 7 ijms-24-01218-f007:**
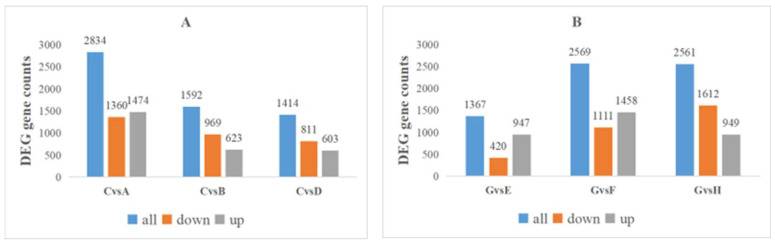
(**A**): The total number of DEGs, up-regulated and down-regulated at 64 DAB; (**B**): the total number of DEGs, up-regulated and down-regulated at 78 DAB.

**Figure 8 ijms-24-01218-f008:**
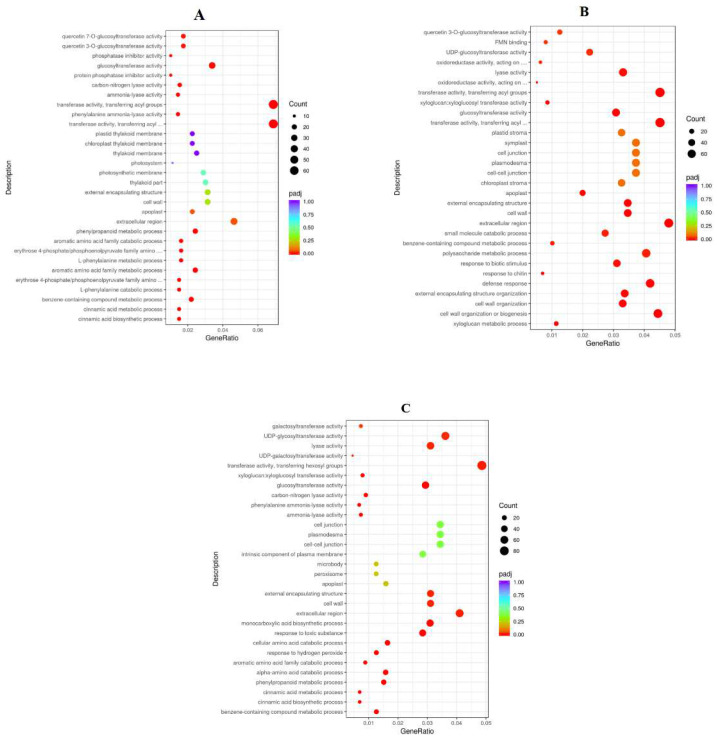
Go enrichment analysis of differentially expressed genes at 78 DAB ((**A**): E vs. G, (**B**): F vs. G, (**C**): H vs. G).

**Figure 9 ijms-24-01218-f009:**
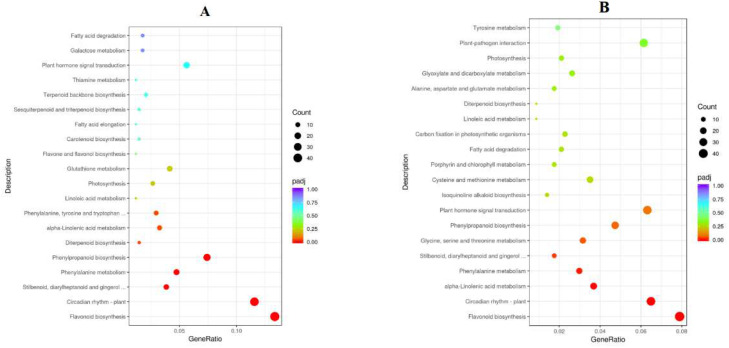
KEGG enrichment analysis of differentially expressed genes at 78 DAB. ((**A**): E vs. G, (**B**): F vs. G, (**C**): H vs. G).

**Figure 10 ijms-24-01218-f010:**
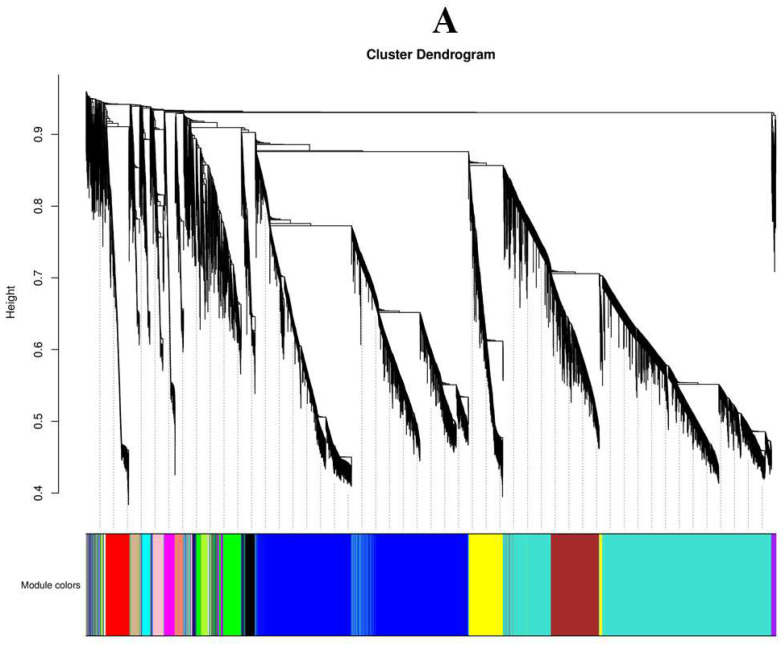
WGCNA identified gene networks and key candidate genes during grape fruit development. (**A**): Hierarchical clusters represent 21 distinct modules with co-expressed genes. Each leaflet in the tree represents an individual gene. (**B**): Pearson correlation-based module trait relationship. The color grids from blue to red represent the r^2^ value (−1 to 1). (**C**–**E**): The red module gene network was significantly correlated with the content of sugar, anthocyanin, and potassium. Key candidates within each network are highlighted in deep red based on highest weight in their respective module and identified for gene descriptors based on annotation results.

**Figure 11 ijms-24-01218-f011:**
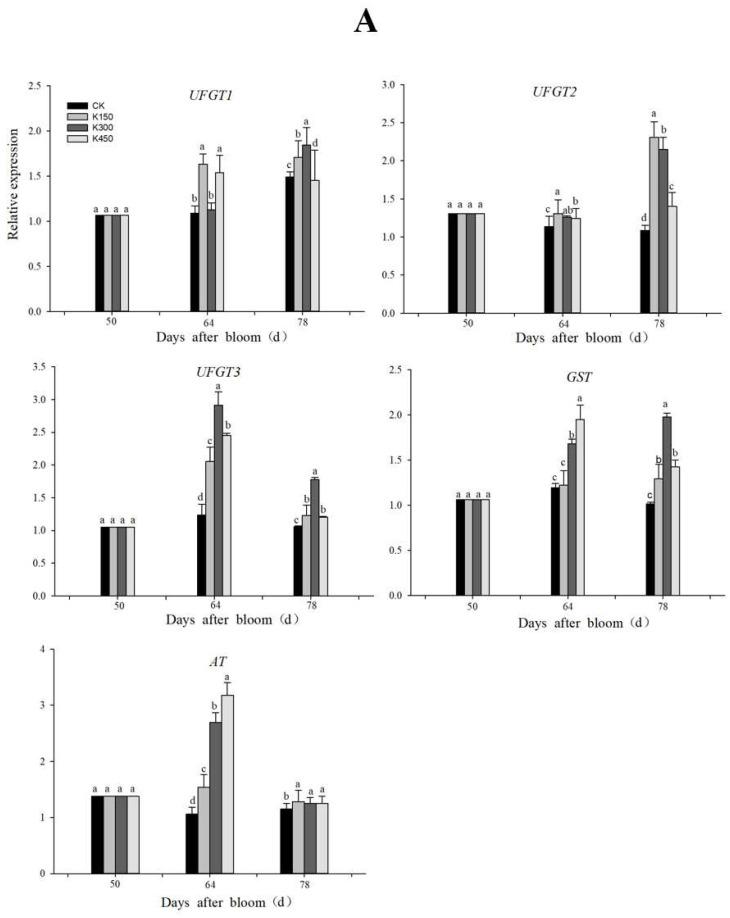
Comparison of expression profiles of representative genes detected by RT-qPCR, different letters (abcd) indicate the difference is significant (Duncan’s Multiple Range Test, *p* < 0.05). The column represents the expression detected by RT-qPCR. (**A**): Relative expression levels of anthocyanin-related genes; (**B**): relative expression levels of sugar-related genes; (**C**): relative expression levels of potassium-related genes.

**Table 1 ijms-24-01218-t001:** Gene primers for RT-qPCR.

Gene Number	Reference Gene Number in NCBI	Forward Primer (5′-3′)	TM (°C)	Reverse Primer (5′-3′)	TM (°C)
VIT_11s0118g00200	XM_002265437	TACTTGGAGGCGATTCTGGA	55.4	GTGGGACTGAATCTCCCTCT	56.5
VIT_03s0063g00050	XM_002285067	CTCGATGTGGAGAAGACAGC	57.5	CCTGAGACCTCGTCAATTCG	57.5
VIT_14s0068g00850	XM_010662357	GAAACCCTTCCCAGCTTTCA	57.5	AAAGTGGTGAAGTGCTCAGG	57.5
VIT_03s0017g00870	XM_010649854	GATGGATTCACTGCTGCTCA	55.4	TGGCAAGAAAGAGGCCAAAT	54.4
VIT_10s0071g01060	XM_010657659	GCCAGTAGTTACTGCCACTC	57.5	CTCTCCACTGAGCATGACAC	57.5
VIT_14s0066g01580	XM_010662640	GCTTGCTGATGAGGAACTGT	55.4	AGCCAGAACAAGCAATACCC	55.4
VIT_07s0104g01730	NM_001280973	TGAGGTGATGGCAGAGAAAG	55.4	TTGTGTGGAATGTCAAATACCTT	54.4
VIT_19s0015g02680	XM_003634701	GCTTCTTGAGATGAACCCGA	57.5	GGTATGGGTCACTAGGCAAC	57.5
VIT_05s0020g03140	NM_001281278	CGTCTCTATATCTTGCGGGC	57.5	CGGTATTAAGCACCACTCCC	57.5
VIT_18s0001g06060	XM_002285734	CCAAACTCAACGACAGAGGT	55.4	ACTGCACATAGGCCATGAAG	55.4
VIT_17s0000g04750	XM_002268053	GGGCGTGTCACATATTGTCT	56.5	GGGGAATTGGGAATGCTAGG	57.5
*VvGAPDH*	GU585870	TTCTCGTTGAGGGCTATTCCA	57.5	CCACAGACTTCATCGGTGACA	57.5

**Table 2 ijms-24-01218-t002:** Potassium content of grape fruit (mg·g^−1^).

Days	Treatment			
	K0-CK	K150	K300	K450
50 DAB	1.135 ± 0.029 a	1.135 ± 0.029 a	1.135 ± 0.029 a	1.135 ± 0.029 a
64 DAB	1.142 ± 0.002 b	1.154 ± 0.006 b	1.256 ± 0.017 a	1.255 ± 0.002 a
78 DAB	1.188 ± 0.013 d	1.332 ± 0.002 b	1.370 ± 0.019 a	1.241 ± 0.020 c

Note: Different letters indicate the difference is significant (Duncan’s Multiple Range Test, *p* < 0.05).

**Table 3 ijms-24-01218-t003:** Soil potassium content (mg·g^−1^).

Days	Treatment			
	K0-CK	K150	K300	K450
78 DAB	27.239 ± 0.156 a	24.184 ± 0.099 d	25.080 ± 0.086 c	25.565 ± 0.166 b

Note: Different letters indicate the difference is significant (Duncan’s Multiple Range Test, *p* < 0.05).

**Table 4 ijms-24-01218-t004:** The berry skin color.

Treatment	*L**	*a**	*b**
K0-CK	23.18 ± 0.20 a	1.27 ± 0.02 b	−1.02 ± 0.14 a
K150	22.43 ± 0.12 b	−1.69 ± 0.06 d	−1.35 ± 0.08 b
K300	21.48 ± 0.16 c	1.00 ± 0.08 c	−1.88 ± 0.09 c
K450	22.41 ± 0.25 b	1.47 ± 0.03 a	−1.48 ± 0.02 b

Note: Different letters indicate the difference is significant (Duncan’s Multiple Range Test, *p* < 0.05).

**Table 5 ijms-24-01218-t005:** Summary of sample sequencing data quality.

Days	Sample	Total Raw Reads	Total Clean Reads	Clean Bases	GC Percentage	Q20 Percentage	Q30 Percentage
50 DAB	T-CK	47158728	45794224	6.87 G	46.93	97.66	93.59
64 DAB	A-CK	46651772	45547160	6.83 G	47.04	97.89	94.11
	B-K150	47439556	45496384	6.82 G	47.12	97.37	92.73
	C-K300	45645408	43878926	6.58 G	47.08	97.71	93.76
	D-K450	46533625	44951584	6.74 G	47.84	97.28	92.85
78 DAB	E-CK	47322815	45934790	6.89 G	47.69	97.42	93.11
	F-K150	49139536	47400020	7.11 G	46.85	97.56	93.38
	G-K300	43948610	42425047	6.37 G	48.40	97.17	92.95
	H-K450	45027932	43493211	6.52 G	47.24	97.56	93.54

**Table 6 ijms-24-01218-t006:** Overview of KEGG analysis of differentially expressed genes.

KEGG ID	Metabolic Pathways	64 DAB			78 DAB			Total Number of Differentially Expressed Genes for Pathway Annotations
		AVSC	BVSC	CVSD	EVSG	FVSG	GVSH	
vvi0480	Glutathione metabolism	15	16	5	14	18	22	137
vvi0052	Galactose metabolism	17	13	9	6	6	15	90
vvi04075	Plant hormone signal transduction	36	20	17	19	36	15	66
vvi0051	Fructose and mannose metabolism	14	7	6	6	7	14	143
vvi00010	Glycolysis/Gluconeogenesis	30	13	22	8	9	19	54
vvi00500	Starch and sucrose metabolism	24	20	16	8	19	19	101
vvi04626	Plant-pathogen interaction	30	15	15	16	35	39	106
vvi00061	Fatty acid biosynthesis	4	4	4	3	7	5	150
vvi00941	Flavonoid biosynthesis	9	4	4	45	45	30	27

**Table 7 ijms-24-01218-t007:** Hub genes selected from three co-expressed modules.

Gene Number	Module	Weight	Annotation
VIT_10s0071g01060	blue	0.3923	Pyruvate kinase (*PK*)
VIT_11s0118g00200	blue	0.4789	Sucrose-phosphate synthase (*SPS*)
VIT_05s0020g03140	yellow	0.5734	Hexose transporter (*HT*)
VIT_03s0063g00050	turquoise	0.5236	Glycosyltransferase (*UFGT1*)
VIT_17s0000g04750	turquoise	0.4572	Glycosyltransferase (*UFGT2*)
VIT_19s0015g02680	turquoise	0.5267	Glutathione-S-transferase (*GST1*)
VIT_14s0068g00850	turquoise	0.5182	Potassium transporter (*KUP1*)
VIT_14s0066g01580	blue	0.4959	Potassium transporter (*KUP2*)
VIT_03s0017g00870	turquoise	0.4123	Anthocyanin acyltransferase (*AT*)
VIT_07s0104g01730	turquoise	0.5097	Potassium transporter (*KUP3*)
VIT_18s0001g06060	turquoise	0.4996	Glycosyltransferase (*UFGT3*)

## Data Availability

Not applicable.

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
