# Peer review of "Identification of Key Genes Induced by Different Potassium Levels Provides Insight into the Formation of Fruit Quality in Grapes"

_ijms, 2023, doi:10.3390/ijms24021218_

Round 1
Reviewer 1 Report (New Reviewer)
The manuscript ‘Identification of key genes induced by different potassium levels provides insight into the formation of fruit quality in grape’ by Huang et al. describes the co-expression of ‘K’ induced genes in the fruit quality of grapes. The manuscript is well-planned, well-executed, and well-written. I have a few observations on the manuscript which may be useful for further improvement;
In title, please mention ‘grapes’ instead of ‘grape’
Please change the red-marked texts into black throughout the text.
Line 18: ‘At 78 DAB’ Please write the full form of DAB on its first appearance.
Line 23: ‘PK expression’ please modify as ‘‘PK gene expressions’
Line 32: ‘Fruit variety’…please delete variety.
Line 34: ‘FAO, 2020’…Please number the reference as [2] and revise the rest of the references accordingly.
Line 41-42: Please delete ‘The same results were obtained in Flame seedless grape’..Please mention the reference [5-6].
Please introduce RNASeq in the introduction section and highlight its importance over other transcriptome studies.
Line 52-57: Please clearly specify the main objective of the study and the future implication of the study briefly.
Line 61-62: Please describe soil conditions and altitude. Please mention the detailed address of the studies institute (City, and country).
Line 83: Please mention the make, city, and country of all the equipment and consumables used in the study.
Line 78, 96, 104: Please remove the years after the citation. Eg. Liu et al. [14]. Please cross-check the journal pattern and revise the citation throughout the text.
Line 112: Please mention the city and country of the company.
Line 113: Please mention the details of the kit used for RNA extraction, purification, and library preparation. Please mention that the work was performed following the manufacturer’s instructions or any specific method followed.
Line 138: Please mention qRT-PCR validation
Line 141: ‘11 genes were selected for RT-qPCR’….Up or down regulated??? Please mention clearly.
Line 144-147: Pl mention the city and country of the company and reagents.
Line 142: ‘Primer 5 software using VvGAPDH’ Pl mention the date of assess of all the software used in the study along with the web link.
Line 142: The primer details for the qRT PCR study may be presented in the main table in stead of the supplementary table. Please mention the Tm against the primers.
Line 150: Please mention ‘Statistical analysis’. Please mention the level of significance here. Please check Spss27.0…or SPSS 27.0? Please write the version of Origin 2021 and re-write the sentence to make it smooth reading.
Table 1: Why no letter appeared against 50 DAB? All units may be written as mg g-1 instead of mg/g
Line 162: Different letters indicate significant differences as per Duncan’s Multiple Range Test (P < 0.05). Please delete line 163.
Figure 1: Please mention the DMRT values (letters) clearly without overlap. Why 50DAB does not contain the values. In Figure 1-4, there are two factors, 1. Days and 2. K-doses. Hence, DMRT may have two factors ‘A, B, C, D’ for days and ‘a.b.c.d’ for K-doses. Please check.
Section 3.6. Please metion the result of l* value. How do you correlate the -ve result of a* value at K150?
Figure 5: The letter size inside the correlation squares may be increased to a visible size. Please mention about the color scheme in the foot note.
I think all the supplementary figures may be presented in the main text as it contains useful information. The top 20 GO depicting different processes may be presented in percentage in a pie diagram for better understanding.
Section 3.9: Please mention the details of the assembly statistics along with the annotation summary, such as final transcripts, total bases, min., and max. sequence length, N50 length, and A+T, and G+C percentage.
Please mention the reference gene(s) taken for qRT PCR study. Why 50 DAB has not been included in the DMRT analysis? It should show a,a,a as there are no significant differences.
The discussion is adequate and justified.
As functional validation has not been carried out in the study, please suggest the need of functional validation to further elucidate the fruit growth mechanisms in grapes owing to K-application/deficit conditions.
Please cross-check the referencing patterns as per the journal standard.
Overall, this is a well-conceptualized, well-presented, and well-written manuscript that contains valuable information and may be considered for publication with minor revision.
Good luck with the revision.
Author Response
Please see the attachment

Reviewer 2 Report (New Reviewer)
Please add the following correction and suggestions before considering the article for publication.
Major comments
1. When the authors were studying the role of potassium in the fruit quality, it is expected that the disease tolerance capacity of K-treated berries against powdery mildew, downy mildew and anthracnose disease would also be taken into consideration, as these diseases causes huge loss in pre-harvest and post-harvest stage, ultimately affecting fruit quality.
2. At different levels of potassium treatment, the concentration of calcium should also be analyzed, as both are interlinked and accumulated at the same point of time.
Round 2
Reviewer 2 Report (New Reviewer)
This manuscript is a resubmission of an earlier submission. The following is a list of the peer review reports and author responses from that submission.
Round 1
Reviewer 1 Report
Dear Authors,
manuscript “Identification of key genes induced by different potassium 2 levels provides insight into the formation of fruit quality in 3 grape” undoubtedly took up an important topic concerning the dependence of the level of potassium fertilization on the quality of grape fruits. To explain these relationships, they used a whole range of methods, including very advanced molecular techniques WGCNA (Weighted Gene Co-Expression Network Analysis) and RT-qPCR. Expression of as many as 7 enzymes involved in metabolism and many biochemical techniques with particular attention to the regulation of the formation of anthocyanins and sugars, but also soluble solids (SSC), and titratable acid (TA) and changes in the concentration of these compounds over time was shown.
Thus, interesting experiments have been carried out and important results have been obtained, but their interpretation does not seem to be correct.
The concentration of K in the soil decreased after the introduction of potassium sulphate and after 78DAB was lower than in the control, but the results, despite statistical significance, differ slightly. Similarly, in grape fruit, the increase in K concentration after its introduction at a dose of K300 is slightly higher than in other K and control tests.
,A tendency to increase the concentration of anthocyanins and sugars with the delivery of K, but only in small doses, has been demonstrated. A particularly interesting relationship is the concentration of two simple sugars: glucose and fructose increasing after the introduction of K and the concentration of tartaric and malic acid falling at the introduction of K.
The authors, however, apparently confused Figures A and mixed sugars with acids in both figures (Fig. 3 and Fig. 4). The expression of genes related to anthocyanins and sugars as well as potassium seems to depend much more on time than on the level introduced externally K.
How do the authors explain these results?
To prove the effect of introduced K on metabolism, authors must perform a correlation analysis as well as a major component analysis (PCA). The results shown only in the form of graphs and the differences shown by means of letter markings of significance are not sufficient to show the relationship between many results obtained.
Legends for Figures should be much more descriptive and provide more information.
The authors did not make a clear hypothesis and did not formulate the objectives of the research. The conclusions drawn must be supported by statistical analyses.
After additional statistical analysis and completion of the above-mentioned important parts of the manuscript, the results will certainly be properly interpreted and the manuscript will gain in value significantly.
Author Response
"Please see the attachment."

Reviewer 2 Report
The authors investigated the effects of exogenously applied four different potassium fertilization levels (0, 150, 300, 450] on the qualities of the grapes. The results showed the increase in sugar and anthocyanin contents, in line with the related gene expressions.
There are several concerns as indicated below.
1) There was less novel point in this study. Since the effects of potassium have been well known in the fruit qualities of grapes so far, the current study has just reconfirmed this. The association of the genes related with anthocyanin biosynthesis, sugar metabolism is easily expected. After RNA-seq analysis, are there any candidate genes related to transcription factors? From the analysis transcription factors, the signal cascade from potassium application to the sugar and anthocyanin accumulation would be uncovered, which is novel study.
2) It has been known that sugar concentrations of grape fruits are different even in one bunch. The authors used 50 berries were randomly removed from each treated fruit. But, the current sample collection method is not clear; therefore, the berries should be collected from the different positions of one bunch and also this sampling method should be applied over several different bunches in the several grapevine trees.
3) To make clearer the effects of potassium, potted grapevine trees (not planted in the orchards) could be suitable for the current purpose.
4) Most parts in the Discussion are the repetition of the Results section.
Author Response
"Please see the attachment."
